# Mindfulness on Rumination in Patients with Depressive Disorder: A Systematic Review and Meta-Analysis of Randomized Controlled Trials

**DOI:** 10.3390/ijerph192316101

**Published:** 2022-12-01

**Authors:** Pan Li, Lingyun Mao, Maorong Hu, Zihang Lu, Xin Yuan, Yanyan Zhang, Zhizhong Hu

**Affiliations:** 1Department of Psychosomatic Medicine, The First Affiliated Hospital of Nanchang University, Nanchang 330006, China; 2School of Public Policy and Administration, Nanchang University, Nanchang 330036, China

**Keywords:** mindfulness-based intervention, depressive rumination, depression, meta-analysis

## Abstract

Objective: To systematically evaluate the effectiveness of mindfulness-based interventions (MBIs) on rumination among patients with depression and their efficacy across countries and year of publication and control conditions. Methods: Web of Science Core Collection, Medline, BIOSIS Citation Index, KCI-Korean Journal Database, SciELO Citation Index, PubMed, Cochrane Library, and Embase were searched to include randomized controlled trials of MBIs for depressive rumination that met the criteria. The Rumination Scale was used as the primary outcome indicator; Depression, mindfulness, and anxiety indexes were selected as the secondary outcome indicators. An evaluation of bias risk was conducted to identify possible sources of bias based on methodological and clinical factors. RevMan5.3 software was used to perform a meta-analysis of the extracted data. Results: Nineteen studies with 1138 patients were included. Meta-analysis showed that MBIs could significantly reduce rumination levels in patients with depression (standardized mean difference (SMD) = −0.46; 95% confidence interval (CI): −0.58, −0.34; *p* < 0.001), notably improve depression (SMD = −0.58; 95% CI: −0.83, −0.32; *p* < 0.001), enhance mindfulness ability (SMD = 0.95; 95% CI: 0.57, 1.32; *p* < 0.001), and reduce the anxiety of patients with depression (SMD = −0.45, 95% CI: −0.62, −0.27; *p* < 0.001). MBIs conducted in Asia improved rumination better than studies in Europe and North America (SMD = −2.05 95% CI: −4.08, −0.01; *p* < 0.001) but had no greater effect than behavior activation on depression. The interventions carried out in the past 5 years were significantly better than earlier studies in improving mindfulness levels (SMD = 2.74; 95% CI: 0.81, 4.66; *p* = 0.005). Conclusions: MBIs are effective in the treatment of depression as they produce pleasant improvement in rumination and depression, decrease the degree of anxiety, and enhance mindfulness levels compared to controls. In newer forms of MBIs, regional differences need to be considered when designing the intervention program. More large, high-quality randomized controlled studies are needed to confirm the conclusion that the effectiveness of MBIs has differences in terms of the trial area and year of publication.

## 1. Introduction

Depressive disorder is one of the most common mental and psychological diseases. Pathogenesis shows that it is affected by biological, psychological, and social factors, and the cognitive vulnerability–stress model emphasizes that negative cognitive style is one of the essential risk factors leading to depressive disorder [1,2]. Ruminative thought, as a maladaptive cognitive style, is implicated in the relationship between biased cognitive processing and mood dysregulation, which is a typical feature of mental disorders [3]; therefore, rumination is regarded as a central mechanism triggering depressive disorders.

Ruminative thoughts cause depressed individuals to think more negatively about the past, present, and future [4]. Response style theory conceptualizes depressive rumination as repetitive thoughts about the symptoms, causes, circumstances, meanings, effects, and consequences of depressed mood and distress.

Rumination exacerbates and prolongs distress, particularly depressive symptoms, through four mechanisms [5,6].

First, rumination is an emotional magnifier that strengthens the effect of depression on thinking, making people more likely to use negative thoughts and memories activated by depression to understand their current situation. For people who are already in a state of depression, rumination can lead to more negative thoughts about the past, present, and future [7]. Rumination has a negative causal effect on emotion and emotion-related cognition in the short term, which will inevitably bring emotional disorders if it occurs for a long time or repeatedly.

Second, rumination interferes with effective problem-solving because it makes thinking more pessimistic and abstract and makes the individual less able to access concrete details of how to solve difficulties.

Third, rumination disrupts instrumental behavior and leads to an increase in stressful environments, such as a reduced willingness to engage in enjoyable activities.

Fourth, it reduces sensitivity to changing contingencies and environments. Studies have shown that rumination impairs attention and central executive function [8], which may prevent patients from responding adaptively to changes in their environment or benefiting from corrective learning that denies negative beliefs [9].

An adaptive and instrumental alternative to confronting depressive rumination is to utilize pleasant or neutral distractions to elevate one’s mood and alleviate depressive symptoms [4]. Mindfulness training is a useful way to interrupt the continuation of disruptive thinking by shifting the individual’s attention to the present moment.

Mindfulness, a state of consciously participating in the present experience with an open, non-judgmental attitude, originates from Buddhist meditation [10] and has been confirmed to play a key role in psychiatric disorders. Behavioral avoidance and cognitive defects caused by a lack of mindfulness are associated with a variety of symptoms (such as depression, compulsion, and self-injury) [11,12]. In response to rumination, mindfulness works through a variety of psychological mechanisms. In the mindfulness process, the individual’s perceptual sensitivity and attention, memory ability, emotional state, and emotional regulation ability will change considerably [13,14,15,16]. Changes in basic cognitive ability modify the primary and advanced processing of internal and external stimuli, which maintain the physical and mental health of individuals, especially those with depression, anxiety, and attention deficits [17]. In addition, one of the remarkable effects of mindfulness is to improve the patient’s mood state and emotion management [18]. Therefore, emotion may also be one of the important psychological mechanisms by which mindfulness plays a role.

Thus, mindfulness-based interventions (MBIs) may work in depressive rumination. MBI is a general term for all kinds of psychotherapy characterized by mindfulness. The more mature ones include mindfulness-based stress reduction (MBSR), mindfulness-based cognitive therapy (MBCT), acceptance and commitment therapy (ACT), and dialectical behavioral therapy (DBT), which have been discussed in theory, supported by a large number of clinical trials in practice, and have a relatively high level of evidence [19,20,21]. Among them, DBT and ACT are outstanding representatives of the third wave of cognitive behavioral therapies (CBTs); an increasing number of psychological practitioners are focusing on and endorsing these two therapies [22,23]. Second, a variety of measures are derived from the interaction of mindfulness with other practices, such as mindfulness-based yoga and mindfulness-based art therapy. A considerable number of interventional studies provide support for these practices [24,25], which are considered to belong to higher empirical levels. However, the negative consequences of rumination may also cause patients to have difficulties in recovering from the psychological disorder and responding to treatment, impair attention and problem-solving skills, reduce responsiveness to external contingencies and feedback, and reduce instrumental action, which may interfere with treatment by limiting the patient’s ability to process ideas or evidence reviewed in psychotherapy or to implement and benefit from a behavioral plan. Some evidence has revealed that the level of rumination at the beginning of treatment influences the effectiveness of CBT interventions for depression [26]. In comparison, rumination at the end of treatment based on mindfulness predicts the recurrence of depression [27], but the effects of MBIs on rumination in depressive disorder and its persistence are not very clear.

The purpose of this study is to summarize and analyze the randomized controlled trials on the effects of MBIs on rumination through meta-analytic techniques and elucidate: (1) whether MBIs are effective in reducing the level of rumination in patients with depression; (2) whether MBIs can effectively reduce the depression level of patients with depression; (3) whether MBIs can enhance the mindfulness ability of patients with depression; and (4) whether MBIs can also improve the anxiety level of patients.

## 2. Method

### 2.1. Search Strategy

The literature was searched from the first available year to September 2022 in the following electronic databases: Web of Science Core Collection, Medline, BIOSIS Citation Index, KCI-Korean Journal Database, SciELO Citation Index, PubMed, Cochrane Library, and Embase.

The search terms were: (mindfulness-based cognitive therapy [Title/Abstract] OR mindfulness-based stress reduction [Title/Abstract] OR acceptance and commitment therapy [Mesh] OR dialectical behavior therapy [Mesh] OR MBCT [Title/Abstract] OR MBSR [Title/Abstract] OR ACT [Title/Abstract] OR DBT [Title/Abstract] OR acceptance-based behavior therapy [Title/Abstract] OR mindfulness-based interventions [Title/Abstract] OR mindfulness-based strategies [Title/Abstract] OR mindfulness-based treatments [Title/Abstract] OR mindfulness-based approaches [Title/Abstract]) AND (rumination [Title/Abstract] OR cognitive rumination [Mesh] OR ruminant thinking [Title/Abstract]).

### 2.2. Inclusion and Exclusion Criteria

Inclusion criteria: (1) study design: randomized controlled trial; (2) study population: patients with at least one episode or remission of depressive disorder diagnosis with depressive or subdepressive symptoms according to clinical criteria; (3) interventions: the experimental group must have received MBIs or techniques that are widely recognized by previous studies, and the control group must have received other interventions, such as CBT or conventional treatment; (4) outcome indicators: the Rumination Scale was used as the primary outcome indicator, and depression, mindfulness, and anxiety indexes were selected as the secondary outcome indicators.

Exclusion criteria: (1) lack of key information and outcome indicators, (2) duplicate published studies, (3) short or incomplete intervention course, and (4) confirmed diagnosis of other severe mental disorders.

### 2.3. Methodological Quality of Studies and Data Extraction

A risk-of-bias evaluation of the final included studies was conducted by two students undertaking a master’s degree in psychology according to the Cochrane Handbook version 5.0.1. The assessment included the following seven aspects: (1) random sequence generation, (2) allocation protocol concealment, (3) blinding of participants and trial personnel, (4) blinding of outcome assessment, (5) completeness of outcome data, (6) selective reporting of results, and (7) other biases. The evaluators judged the seven aspects as “low bias”, “high bias”, and “unclear”, and a third evaluator (also a student undertaking a master’s degree in psychology) was consulted to discuss the evaluation results in case of disputes.

The following data were extracted from the included articles: (1) the first author, year of publication, and nationality of the articles; (2) the sample size, gender, and age of the experimental and control groups; (3) the interventions, intervention duration, and follow-up time of the experimental and control groups; and (4) the mean (*M*) and standard deviation (SD) used for the outcome indicators.

### 2.4. Statistical Analysis

RevMan 5.3 software was used to process the analysis. In meta-analyses, the fixed-effects model assumes that the studies are homogeneous, whereas the random-effects model assumes heterogeneity between studies [28]. *I*^2^ statistics reflect the proportion of the heterogeneity component in the total variance of the effect size. *I*^2^ > 50% indicates relatively substantial heterogeneity [29]. A fixed-effects model was selected if *p* > 0.1 and *I*^2^ < 50%, and a random-effects model was selected if *p* < 0.1 and *I*^2^ > 50%. Sensitivity analysis or subgroup analysis was used to explore the sources of heterogeneity and the effects on the results. The measures are expressed as weighted mean difference (MD) with a 95% confidence interval (95% CI), and *p* < 0.05 indicates that the difference is statistically significant. If different scales were used for the same outcome indicator, standardized mean difference (SMD) was used in the analysis. The value of SMD represents the difference in the effect of the experimental group compared with the control group. Larger SMD values represent a greater difference in the outcome of the two intervention groups, and symbols refer to the strengthening or weakening of the outcome indicators after the intervention. The CI that follows indicates whether the difference is statistically significant; two values containing zero in between imply that the difference is not significant.

In the forest plot, the square represents the point estimate of the study effect size, the size indicates the weight of each study, and the straight line extending from both sides of the square represents the CI of the effect size. The longer the line, the wider the CI and the less precise the result. The vertical line is the null line. If the CI of the study intersects with the null line, then the study effect size is not statistically significant, which corresponds to the CI spanning zero. The diamond represents the merger effect size, the center of gravity of the diamond is the point estimate of the merger effect, and the width is the CI of the merger effect size.

## 3. Results

### 3.1. Article Screening Process and Results

A total of 1816 articles were obtained, and 731 articles remained after eliminating duplicate articles by screening by type of literature and journal of publication. A total of 571 articles were excluded by reading the titles and abstracts of the articles, and 19 randomized controlled studies were finally included after reading the full text. The process of literature inclusion is shown in Figure 1.

### 3.2. Basic Characteristics of Included Articles

The 19 studies [30,31,32,33,34,35,36,37,38,39,40,41,42,43,44,45,46,47,48] included were all focused on MBIs, and the specific intervention programs were diversified in terms of group intervention. Females made up more than 50% of all the studies. The control groups had different measures. Five studies used behavioral activation, including behavioral activation modeled from the Brief Behavioral Activation Treatment for Depression [35,41], physical exercise guided by a physical trainer [46], a walking control condition [39], and cognitive behavior therapy [45]. Three studies reported on placebo control, including relaxation [31], pill placebo [34], and psychology education [37]. Eleven studies used blank controls (BCs), including waiting lists [30,33,36,40,47,48] and treat-as-usual methods [32,38,42,43,44]. Five articles reported follow-up data with a follow-up period of 1–6 months. In addition, six studies reported that MBIs were better than BCs. Five studies reported no remarkable differences between MBIs and BCs. Two studies reported that MBIs were better than behavior activation (BA). Two studies reported no remarkable differences between MBIs and BA. Two studies reported that MBIs were better than pseudostimulus. Two studies reported no substantial differences between MBIs and placebo control. The basic characteristics of the included studies are shown in Table 1.

### 3.3. Risk of Bias of Included Studies

The included articles were all RCTs, and the experimental groups and control groups had no remarkable differences at baseline. Seventeen studies reported the reasons and numbers of participants who dropped out, and five studies were blinded to outcome assessments. A double-blind design was not possible because of the specificity of psychotherapy, and the risk of performance bias was high in all studies. Sixteen studies described the method of random sequence generation, and fifteen of them mentioned allocation concealment. Other risks of bias were unclear. The risk-of-bias report is summarized in Figure 2, where “+” means “low bias”, “−” means “high bias”, and “?” means “unclear”. Overall, the included literature has some risk of bias, but each study is still acceptable.

### 3.4. Meta-Analysis

#### 3.4.1. Effects of MBIs on Rumination in Patients with Depressive Disorder

All 19 studies reported the effect of MBIs on the rumination of patients who were diagnosed with depression. Figure 3 demonstrates a random-effect meta-analysis comparing mindfulness and control treatment on participants’ rumination symptoms at postintervention intervals. The figure reveals a relatively low level of heterogeneity among the 19 studies of 1138 participants (*I*^2^ = 37%; *n* = 19). Two studies [35,38] indicated the negative effects of mindfulness over control treatment, with no statistically substantial difference. Regarding the pooled data in the meta-analysis, the results show the pooled effect size of mindfulness on patients’ rumination, with an SMD of *g* = −0.46 (95% CI = −0.58–−0.34) and a calculated effect size of *Z* = 7.53 at *p* < 0.001. The overall effect was favorable for MBIs and was statistically significant, which suggested that MBIs have a definite positive effect overall in the treatment of rumination in patients with depression, regardless of which method was used in the comparison.

For heterogeneity, a subgroup analysis was conducted by classifying all included studies into three categories in accordance with the area of trials: ten studies with 853 participants were implemented in Europe, five studies with 165 participants were conducted in North America, and three studies with 75 participants were conducted in Asia. One study in Oceania was excluded for not meet the criteria of subgroup. Heterogeneity was low in Europe (*I*^2^ = 0%, *n* = 10) and North America (*I*^2^ = 0%, *n* = 5) but relatively high in Asia (*I*^2^ = 84%, *n* = 3). The pooled data were SMD of *g* = −0.46 (95% CI = −0.60–−0.33) and *Z* = 6.64 at *p* < 0.001 for Europe, SMD of *g* = −0.30 (95% CI = −0.60–0.01) and *Z* = 1.89 at *p* < 0.001 for North America, and SMD of *g* = −2.05 (95% CI = −4.08–−0.01) and *Z* = 1.97 at *p* = 0.05 for Asia. The effect of all three areas is enlightening. Subgroup analyses showed no significant difference between subgroups (*I*^2^ = 40.3%, *n* = 3). Further analysis compared the subgroup differences between Europe and Asia. The results showed statistically significant differences between the subgroups of Europe and Asia (*I*^2^ = 57.0%, *n* = 2) and between those of North America and Asia (*I*^2^ = 64.1%, *n* = 2). The results indicate that the area of trials was one of the sources of heterogeneity in this subgroup. The MBIs for rumination for patients with depression implemented in Asia were considerably more effective than those implemented in Europe and North America, but the effects of MBIs in Asia were similar to those in Europe and North America (Figure 4 and Figure 5).

#### 3.4.2. Effects of MBIs on Mindfulness in Patients with Depressive Disorder

Thirteen studies of 891 participants investigated the effects of MBIs on mindfulness with high heterogeneity (*I*^2^ = 83%, *n* = 13). Pooled data in the meta-analysis demonstrated the significant impact of MBIs on patients’ mindfulness with an SMD of *g* = 0.95 (95% CI = 0.57–1.32) and *Z* = 4.98 at *p* < 0.001.

The studies were divided into three subgroups according to the year of publication. Four studies with 180 participants were published between 2017 and 2022, six studies with 416 participants were published between 2012 and 2016, and three studies with 295 participants were published between 2007 and 2011. The heterogeneity was relatively high in the period of 2017–2022 (*I*^2^ = 95%, *n* = 4) and low in the periods of 2012–2016 (*I*^2^ = 0%, *n* = 6) and 2007–2011 (*I*^2^ = 30%, *n* = 3). The pooled data showed an SMD of *g* = 2.74 (95% CI = 0.81–4.66) and *Z* = 2.78 at *p* = 0.005 for 2017–2022, an SMD of *g* = 0.69 (95% CI = 0.50–0.89) and *Z* = 6.84 at *p* < 0.001 for 2012–2016, and an SMD of *g* = 0.59 (95% CI = 0.27–0.91) and *Z* = 3.64 at *p* < 0.001 for 2007–2011. The efficacy of MBIs increased with the year of publication. The heterogeneity between subgroups (*I*^2^ = 57.1%, *n* = 3) shows that the year of publication was one of the sources of variation, and the level of mindfulness of patients with depression also increased partially after MBIs. The treatment effects of MBIs implemented in recent years have been more advantageous compared with past interventions (Figure 6).

#### 3.4.3. Effects of MBIs on Depression in Patients with Depressive Disorder

Depression was measured in 18 of the reviewed trials. The results show a relatively high level of statistical heterogeneity (*I*^2^ = 71%, *n* = 18) among these 18 studies of 1108 subjects. Sensitivity analysis revealed that the *I*^2^ statistic decreased (*I*^2^ = 53%, *n* = 16) after excluding two studies [30,38]. Fifteen of these studies showed the positive effects of the mindfulness groups over the control groups, with pooled data of an SMD of *g* = −0.58 [95% CI = −0.83–−0.32] and an effect size of Z = 4.47 at *p* < 0.001. These values demonstrate the remarkable impact of mindfulness on patients’ depressive symptoms.

The studies were divided into two subgroups, namely, BC and BA, according to the interventions of the control group. Two unclassifiable studies were excluded from the analysis. Subgroup analysis showed a low level of heterogeneity between studies in the BC group (*I*^2^ = 0%, *n* = 8). MBIs relieved depression symptoms in patients according to the pooled data, with an SMD of *g* = −0.59 (95% CI = −0.74–−0.45) and an effect size of *Z* = 8.01 at *p* < 0.001. Heterogeneity between studies in the BA group was also low (*I*^2^ = 29%, *n* = 5). Random-effects meta-analysis for comparing MBIs and other interventions on participants’ depressive symptoms showed a pooled effect size of an SMD of *g* = −0.13 (95% CI = −0.51–0.24) and *Z* = 0.68 at *p* = 0.50. This result suggests an unremarkable favorable trend for BA instead of MBIs, and the overall effect was supportive of MBIs. Heterogeneity was found between the subgroups (*I*^2^ = 80.3%, *n* = 2), indicating that the type of control group was one of the sources of heterogeneity in this subgroup. The effect of MBIs on depression improvement in patients was considerably higher than those of other methods in general, and the advantage of MBIs over the BC group was further expanded. However, the effect of MBIs on depression was the same, relative to the BA group (Figure 7).

#### 3.4.4. Effects of MBIs on Anxiety in Patients with Depressive Disorder

Nine studies with 790 subjects were included, and no statistical differences were found between studies (*I*^2^ = 26%, *n* = 9). The pooled effect size of mindfulness on patients’ anxiety with an SMD of *g* = −0.45 (95% CI = −0.62–−0.27) and an effect size of *Z* = 5.05 at *p* < 0.001. The overall effect was significant and favored MBIs. MBIs are also better at reducing the level of anxiety than other measures (Figure 8).

#### 3.4.5. Analysis of Article Publication Bias

Publication bias refers to the fact that study results with statistically significant findings are more likely to be reported and published than non-significant and invalid results. Publication bias analysis is performed through a funnel plot. The horizontal coordinate is the effect size of the original study, and the vertical coordinate is the sample size, standard error, or precision of the original study. The larger the sample size, the more concentrated the distribution. The graph is symmetrical and funnel-shaped if the study has no bias.

With regard to the funnel plot, the left−right distribution of the plot was slightly asymmetrical, and two studies were far from the center. The results show the main sources of heterogeneity between the studies, one of which is due to the high rate of loss, while the other has a large difference in the baseline level, suggesting the possibility of publication bias. Although the majority of studies were clustered at the top and more evenly distributed on both sides, the overall risk of publication bias remained at a favorable level (Figure 9).

## 4. Discussion

In the concept of mindfulness, the failure to consciously pay attention to present experiences and the rejection of existing experiences are important causes of psychological pain. Lack of focus on the present leads to unpleasant emotions, and the ability to focus is associated with higher happiness in daily life. For most patients with depression, rumination becomes a fixed negative cognitive model. An annoyed mindset is maintained; thus, individuals fall into thinking frequently and ignore the real experience that they are undergoing. Rumination causes patients to focus on displeasure and urges them to think about the causes or effects of depression [5], which substitutes a negative attitude towards events that have happened. As a result, individuals have an evasive and hateful mind towards the past and related things. Mindfulness, through a series of means, such as acceptance and awareness and attention to the present, fosters a willingness to live with all thoughts, attitudes, memories, and emotions. In this way, awareness and control over attention are strengthened, which further lowers the degree of repeated thinking and the resulting emotional reactions. Patients can then better devote themselves to their daily life and pursue their own goals.

Previous literature has reported the benefits of MBCT on rumination for patients with depressive disorders [49]. This meta-analysis provides a wider range of clinical therapy in evaluating the effectiveness of MBIs on patients with depression. We identified 19 RCTs that assessed the effectiveness of MBIs versus BC (TAU + WL) on the rumination, depression, mindfulness, and anxiety of patients diagnosed with depressive disorder. Some heterogeneity exists between these studies (e.g., area of trials, control group intervention, year of publication), but the results are statistically consistent.

The meta-analysis results showed that MBIs had a certain effect on patients with depression. MBIs decreased the rumination level at the end of the intervention. State rumination and trait rumination will affect the individual’s ability to shift attention, interfering with task completion [50]. By observing one’s own thoughts without judgment, mindfulness successfully distances oneself from the content of rumination. The practice of mindfulness enhances awareness of the present, and this cognitive change, brought about by self-observation, has a long-term effect [51]. Mindful skills will also be steadily developed over the course of treatment, which may help maintain and further improve efficacy during the follow-up period.

The depression of patients was also relieved after treatment. Notably, half of the included studies found the relief to be unremarkable, possibly because mindfulness does not consider changes in mood as the target of intervention. Mindfulness is concerned with how to interrupt people’s unconscious thoughts in daily life so that attention and consciousness are based on a person’s present experience, which can take many forms, including physical sensation, emotional response, psychological image, psychological conversation, and perceptual experience [10]. This monitoring feature is described as “observing” or “conscious of each presented experience” [52]. Second, adopting an open or receptive attitude toward these experiences, including paying attention to experiences in a curious, detached, and non-reactive style, is essential. Mindfulness emphasizes active and conscious attention to personal experience, whether the experience is positive or not. This concept helps people find a fulcrum in complicated environments and runaway thinking. A large number of studies have confirmed that mindfulness can improve individual happiness, life satisfaction, life meaning, and other positive aspects [53,54]. Therefore, the change in symptoms may be a by-product of mindfulness therapy. Previous studies have also confirmed that mindfulness does not directly cause changes in depression but regulates mood by influencing intermediary factors [55]. The improvement in mindfulness levels was also verified by meta-analysis.

In addition, the results of the meta-analysis show that MBIs can improve the anxiety of patients with depression. Notably, a corresponding relationship exists between worry, a cognitive component of anxiety, and rumination. Worry refers to the repeated thinking and negative prediction of unknown events by individuals in the form of words [56]. Worry and rumination are similar but different. Both of them are called repetitive negative thinking and can predict pathological symptoms. The difference is that worry is a kind of thinking about events that may happen but are not yet happening, pointing to the future, whereas rumination is a review of events that have been experienced, pointing to the past. The two modes of thinking are a cross-diagnostic process and may cause multiple co-diseases [57]. Thus, the same therapy may be used to treat multiple mood disorders clinically and to gain more benefits from cross-diagnostic programs that address the core pathological processes of repetitive negative thinking (rumination and worry) related to the disease. However, more research is needed to verify this assumption in the practical aspect.

The results of the subgroup analysis showed that the publication time of the study affected the intervention effect of MBIs on the mindfulness level of patients with depression and that the curative effect gradually increased over time. This finding may be because the publication time of the study represents the level of MBIs at that time; mindful ideas have been in continuous development in the past two decades. The development of its concept, form, and practice has also promoted the birth of the third wave of therapy [23]. Advances in scientific research have caused mindfulness to gradually enter the lives of ordinary people. People know more about mindfulness, and its acceptance has also gradually increased. Notably, although the results of the subgroup analysis showed that later studies were more effective than earlier studies on the improvement of mindfulness in patients with depressive disorder, this conclusion carries some risk and needs to be viewed with caution because of the inclusion of literature from studies in the last decade.

The subgroup analysis results identified that the different areas of studies had an influence on the rumination level of patients with depression, and the effect of MBIs in Asia was much better than those in Europe and North America. The subgroup analysis also showed that the mindfulness level of patients with depression in Asia improved the most after the intervention, which may be related to the differences in regional cultures. Buddhism is the source of mindfulness, and it originated in Asia. The concept of mindfulness is still widespread in a considerable number of countries, nationalities, and religions in Asia and is a part of their culture.

Subgroup analysis of depression levels among different control groups showed that MBIs had a better therapeutic effect on depression than BA and BC, but MBIs had no statistically significant improvement in depression levels compared with BA. Mindfulness in the treatment of patients with depression was affirmed. To a certain extent, the use of mindfulness in mood improvement has the same effect as BA.

In summary, MBIs, as an effective practice, are qualified in clinical psychotherapy for patients with depression, and the design of MBIs should be carried out with the latest forms and theories of MBIs as references. We need to take regional differences into account, pay attention to the acceptance of MBIs in different regions, base the design on local characteristics, fully discern the advantages and disadvantages, and choose the treatment plan according to the situation, which will help mindfulness to be better applied in the treatment of depressive disorders.

## 5. Conclusions

This systematic review confirmed the efficacy of MBIs in improving rumination, depression, mindfulness, and anxiety in patients with depressive disorders but was insufficient to demonstrate that MBIs enhance the efficacy of depression during follow-up. The inclusion of data from 11 countries worldwide further enhances the generalizability of the results. However, the current research has the following limitations. (1) The number of included studies was small, and the included studies had differences in intervention forms (MBCT, MBSR, ACT, etc.), the type of control group, the use of scales, reporting methods, and intervention cycles, which may lead to heterogeneity among studies; the possible impact of these difference is not clear. (2) A small *N* was present in a portion of the included articles, which may have caused the obtained effect values to deviate from the actual effect. This factor needs to be considered and further confirmed in future studies. (3) The literature included in the study did not mention whether to carry out a double-blind design. Most of the studies did not explain the distribution concealment, and some asymmetry was observed in the funnel map distribution of mindfulness-related interventions on the rumination of patients with depression, suggesting the possible presence of publication bias. The results of the study need to be carefully explained, and future studies are required to improve the quality of research. (4) Some studies were excluded because their data could not be extracted, which may affect the analysis results to some extent. A follow-up study can explore the similarities and differences between different treatments of mindfulness in the intervention of depressive disorders and select more economical, simple, easy-to-implement, and effective measures for the clinical treatment of patients with depression.

## Figures and Tables

**Figure 1 ijerph-19-16101-f001:**
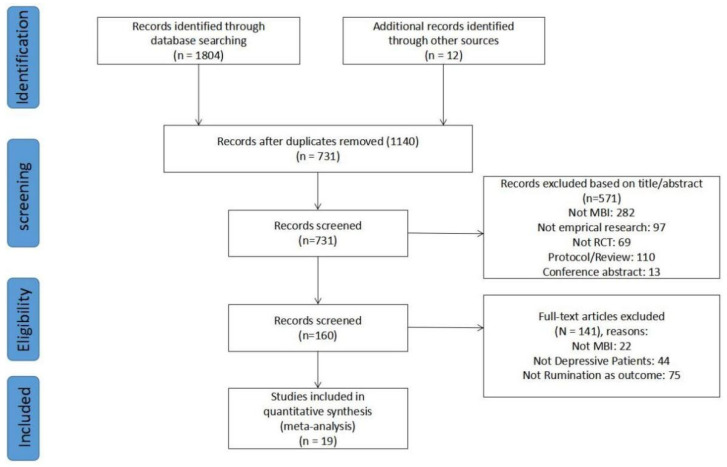
Flow chart of the study selection process.

**Figure 2 ijerph-19-16101-f002:**
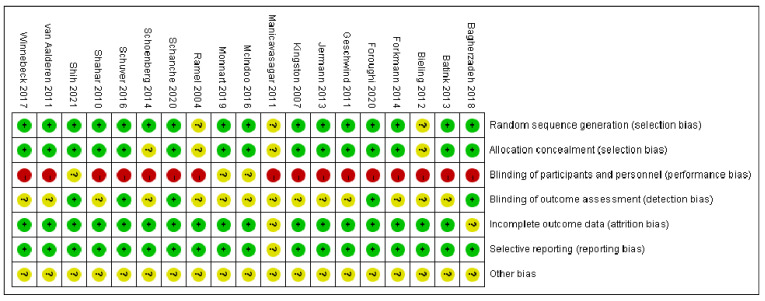
Risk of bias of individual studies [30,31,32,33,34,35,36,37,38,39,40,41,42,43,44,45,46,47,48].

**Figure 3 ijerph-19-16101-f003:**
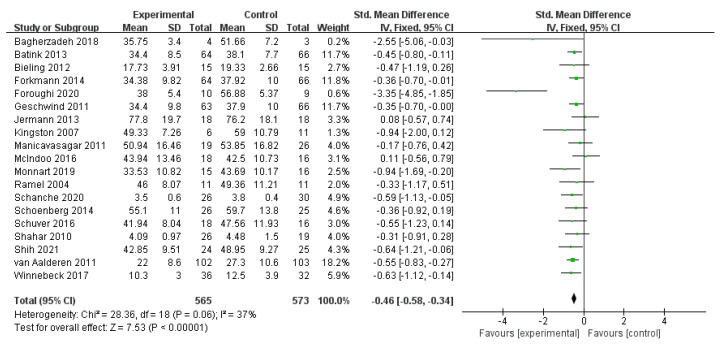
Meta−analyses of the effects of MBIs on rumination [30,31,32,33,34,35,36,37,38,39,40,41,42,43,44,45,46,47,48].

**Figure 4 ijerph-19-16101-f004:**
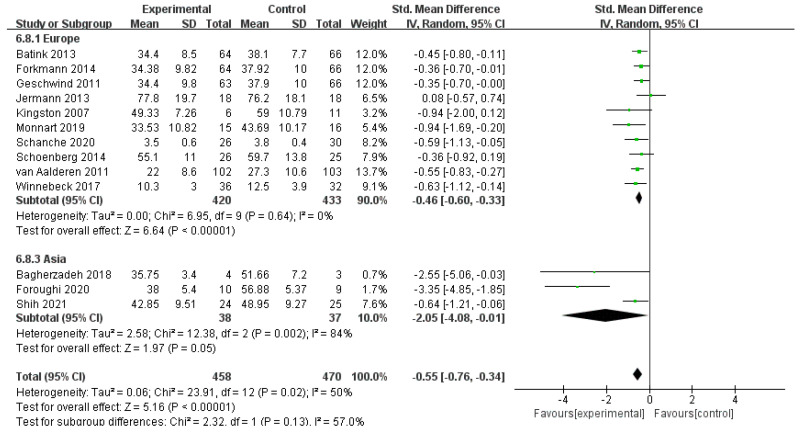
Meta−analyses of the effects of MBIs on rumination between Europe and Asia [30,31,32,36,37,38,40,41,42,43,44,46,48].

**Figure 5 ijerph-19-16101-f005:**
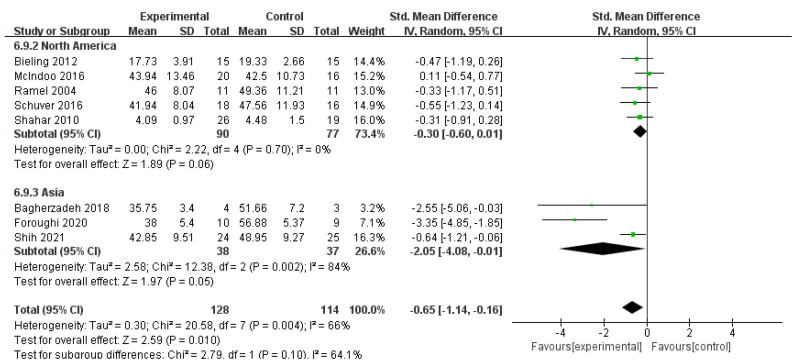
Meta−analyses of the effects of MBIs on rumination between North America and Asia [30,33,34,35,39,41,46,47].

**Figure 6 ijerph-19-16101-f006:**
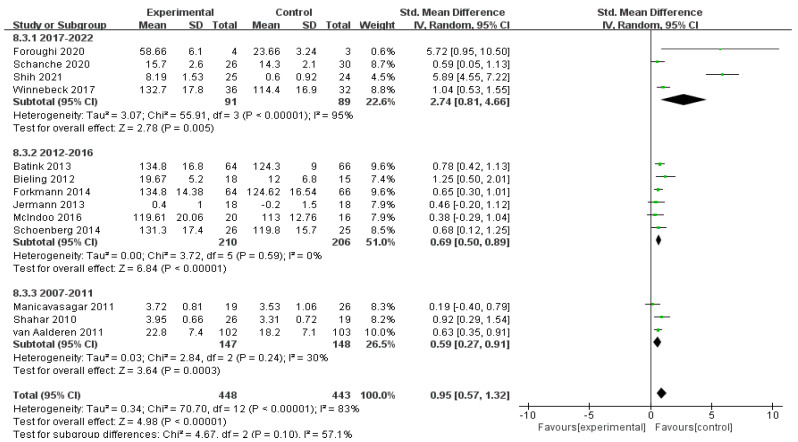
Meta−analyses of the effects of MBIs on mindfulness between years of publication [30,31,32,33,34,35,36,37,38,43,44,45,46,48].

**Figure 7 ijerph-19-16101-f007:**
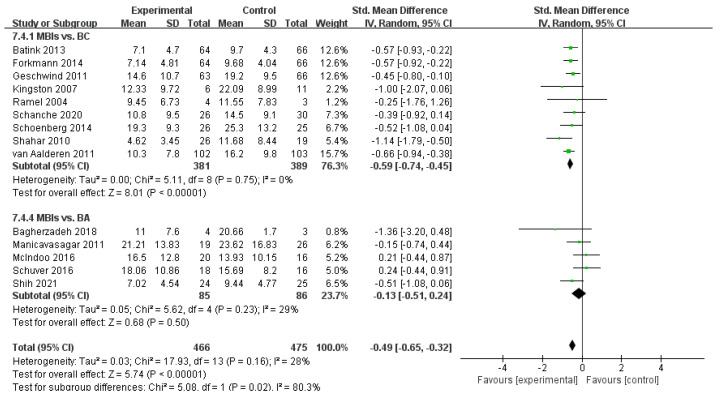
Meta−analyses of the effects of MBIs on depression between interventions of the control groups [32,33,35,36,39,40,41,42,43,44,45,46,47,48].

**Figure 8 ijerph-19-16101-f008:**
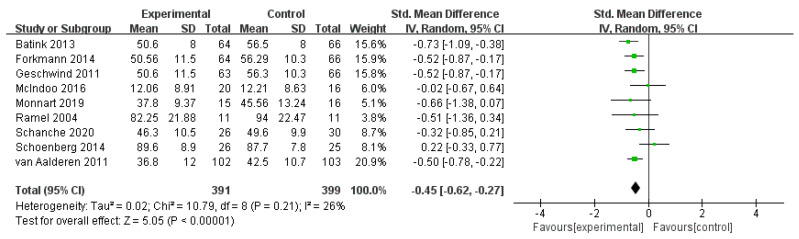
Meta−analyses of the effects of MBIs on anxiety [31,32,35,36,40,43,44,47,48].

**Figure 9 ijerph-19-16101-f009:**
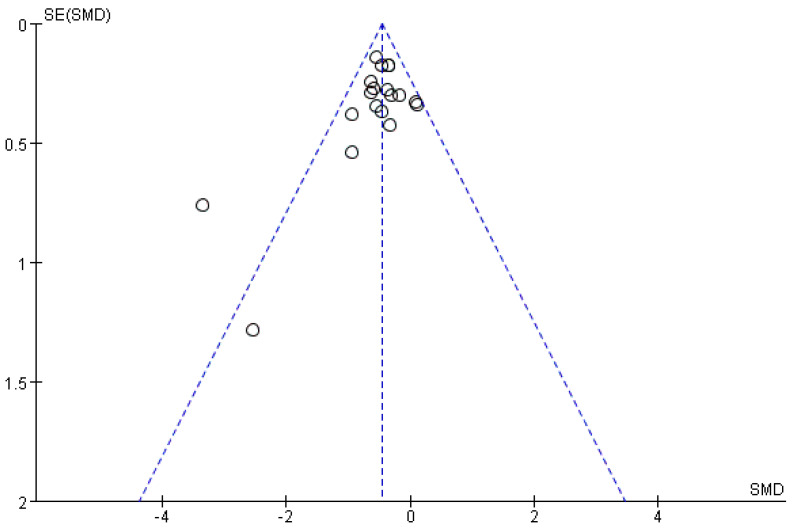
Funnel plot of rumination.

**Table 1 ijerph-19-16101-t001:** Characteristics of the nineteen studies included.

Study	Country	N	Female	Intervention Measures		Follow-Up Time (Months)	Ex vs. Con	Outcome Measure			
Author, Year		Ex/Con	%	Ex	Con			Rumination	Depression	Mindfulness	Anxiety
Foroughi et al. [30], 2020	Iran	10/9	72.2	MBCT	WL	1	>	RRS	BDI-II	SMQ	/
Monnart et al. [31], 2019	Belgium	15/16	64.5	MBCT	relaxation	/	>	SARI	MADRS	/	STAI
Batink et al. [32], 2013	Netherlands	64/66	75.4	MBCT	TAU	/	>	RSS	HDRS	KIMS	PSWQ
Shahar et al. [33], 2010	USA	26/19	84.4	MBCT	WL	/	=	RRS	BDI	MAAS	/
Bieling et al. [34], 2012	Canada	15/15	58	MBCT	pill placebo	/	=	EQ-R	/	TMS	/
McIndoo et al. [35], 2016	USA	18/12	58.8	MBT	BA	1	=	RRS	BDI-II	FFMQ	BAI
Schanche et al. [36], 2020	Norway	26/30	73.4	MBCT	WL	/	>	RRQ	BDI-II	FFMQ	STAI
Winnebeck et al. [37], 2017	Germany	36/32	60.3	MBI	psycho education	/	>	RRS	BDI-II	FFMQ	/
Jermann et al. [38], 2013	switzerland	18/18	69.4	MBCT	TAU	6	=	RRQ	BDI-II	MAAS	/
Schuver et al. [39], 2016	USA	18/16	100	MBYI	walking control	1	=	RSS	BDI	/	/
Geschwind et al. [40], 2011	Netherlands	63/66	75.6	MBCT	WL	/	>	RRS	IDS	/	STAI
Bagherzadeh et al. [41], 2018	Iran	4/3	85.7	ACT	BA	1.5	>	RRS	BDI-II	/	/
Kingston et al. [42], 2007	Ireland	6/11	88.2	MBCT	TAU	/	=	RRS	BDI	/	/
Forkmann et al. [43], 2014	Germany	64/66	75.4	MBCT	TAU	/	>	RSS	HDRS	KIMS	PSWQ
van Aalderen et al. [44], 2011	Netherlands	102/103	70.7	MBCT	TAU	/	>	RSS	BDI	KIMS	PSWQ
Manicavasagar et al. [45], 2011	Australian	19/26	64.4	MBCT	CBT	/	=	RRS	BDI-II	MAAS	/
Shih et al. [46], 2021	China	24/25	87.7	MBCT	BA	/	>	RRS	HDRS	MAAS	/
Ramel et al. [47], 2004	USA	11/11	45.5	MBSR	WL	/	=	RSQ	BDI	/	STAI
Schoenberg et al. [48], 2014	Netherlands	26/25	62.8	MBCT	WL	/	=	RRS	IDS	FFMQ	STAI

Note. Ex = experimental; Con = control; MBCT = mindfulness-based cognitive therapy; MBT = mindfulness-based therapy; MBI = mindfulness-based intervention; MBYI = mindfulness-based yoga intervention; ACT = acceptance and commitment therapy; MBSR = mindfulness-based stress reduction; WL = waiting list; TAU = treat-as-usual; BATD = behavioral activation; CBT = cognitive behavior therapy; RRS = Ruminative Response Scale; SARI = Sadness and Anger Ruminative Inventory; RSS = Rumination Response Style Questionnaire; EQ-R = Experiences Questionnaire-Rumination scale; RRQ = Rumination–Reflection Questionnaire; BDI-II = Beck’s Depression Inventory-II; MADRS = Montgomery–Asberg Depression Rating Scale; HDRS = Hamilton Depression Rating Scale; IDS = Inventory of Depressive Symptoms; SMQ = Southampton Mindfulness Questionnaire; KIMS = Kentucky Inventory of Mindfulness Skills; MAAS = Mindful Attention Awareness Scale; FFMQ = Five Facet Mindfulness Questionnaire; TMS = Toronto Mindfulness Scale; STAI = State-Trait Anxiety Inventory; PSWQ = Penn State Worry Questionnaire; BAI = Beck’s Anxiety Inventory.

## Data Availability

All data used in the study are presented in the graphs of the article, and also available by reviewing the cited literature of included studies.

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
