# Peer review of "Mindfulness on Rumination in Patients with Depressive Disorder: A Systematic Review and Meta-Analysis of Randomized Controlled Trials"

_ijerph, 2022, doi:10.3390/ijerph192316101_

Round 1

Reviewer 1 Report

Hello - thanks for the opportunity to review this article. It represents an interesting meta-analysis on a very important topic - consideration of the issues of "mindfulness intervention" and "Ruminating". While the topic is very relevant, there are issues related to the concept of mindfulness that need further discussion.

Please see the detailed comments below for specific issues:

Introduction

Line 42 and 43. At the beginning of the introduction there is a discrepancy, as if some text had been deleted. Is it possible that this was a mistake in the editing?

Line 73.  I am not sure what is meant in the sentence "Mindfulness, a state of consciously participating in the present experience with an open, non-judgmental attitude, originates from Buddhist meditation[10], has been confirmed to play a key role in psychiatric disorders" Do you mean mindfulness or mindfulness intervention? I do not understand the role mindfulness can play in the course of a psychiatric disorder.

Line 85. For the first time the term "mindfullness intervention" is introduced. I believe that this concept is much more complex than it appears here and should be explained. There are different types of interventions and each of them have different levels of evidence. However, the authors do not differentiate between them and do not enter into this debate, which I believe is necessary.

Line 14, 85, 95, 98, 123, 205... The abbreviation MBIs is often not used properly. In some cases the term mindfulness intervention is used, but I do not know if it is the same as MBIs. These details should be corrected.

Results

Table 1. This table specifies the type of MBIs used in each study. However, in one of them it is not specified and MBI appears (Winnebeck et al, 2017).

Table 1.  The small N of some of the studies that have been included in the meta-analysis is surprising. Has this not been taken into consideration as an exclusion criterion?

Discussion and conclusion

Aunque se menciona la variabilidad de MBIs de los estudios seleccionados  como una limitación, pienso que este sesgo debe ser discutido en mayor profundidad.

References

Reference 34 BAGHERZADEH LEDARI R, MASJEDI A, BAKHTYARI M, et al. 2018 is listed as IN PRESS      

Reviewer 2 Report

Mindfulness on Rumination in Depressive Patients: A Systematic Review and Meta-Analysis of Randomized Controlled Trials

Summary and Strengths: The goal of this paper was to systematically evaluate the effectiveness of mindfulness-based interventions for patients with depression. Overall, the authors appear to follow gold-standard guidelines for conducting a systematic review and meta-analysis. The statistical methods and reporting of results appear to be appropriate. The scope of the topic is also important should be of interest to many readers of I-JERPH. I commend the authors for reviewing studies through September 2022, which will help this article be as up-to-date as possible upon publication. Below, I detail several potential areas for improvement.

Major:

-        The authors would benefit from further proof-reading this article and correcting numerous grammatical mistakes. It interferes with the reader’s experience, and during the introduction, can potentially hinder the reader’s understanding of the literature and rationale for the study. While I commend the authors for publishing in what is assumed to be their non-native language, I encourage them to seek consultation from a native English speaker/translator who can provide additional edits throughout.

-        The scope of the research question is not clear from the abstract. The authors first state the Objective: “to systematically evaluate the effectiveness of mindfulness-based interventions (MBIs) for patients with depression.” However, it seems that this is not true, as the primary outcome measure is rumination. So, is the primary objective to evaluate the effects of MBIs on rumination among patients diagnosed with depression? This must be made clear in the abstract.

-        The authors note that MBIs did not perform better than behavioral activation in Asia, but this research question is not set up in earlier in the abstract. It would help to know that the authors planned to compare MBI efficacy across countries (I assume this wasn’t a post hoc analysis). This is the first mention of behavioral activation as well. Was the used as a control intervention in multiple studies, or all of them? It’s difficult to tell whether MBIs performed better than BA just in Asia or whether this comparison can be made in other countries.

-        As I note below, transparency on the nature of the control conditions of the included studies is a broader concern. Understanding what types of control conditions were included can greatly help readers interpret these results.

Minor comments:

-        Throughout the manuscript, I would encourage the authors to use more modern and less stigmatizing language, such as “patients with depression” instead of “depressive patients.”

-        Line 171: What is “pseudo stimulus”? What is “blank control”? Is this a typo?

-        Table 1: There is a lot of information here, which is mostly useful, but the full country name should fit on one line, or else the formatting looks unprofessional

-        Somewhere in the results, it would be helpful to understand the nature of the control conditions in the included studies. Were these waitlist controls or something more active? This helps the reader interpret the overall effects.

-        Trials is consistently misspelled as “trails.”

-        In the discussion section, the authors should be careful to not conflate lack of focus with rumination. At least, they should take more care in bridging the gap from the mindfulness (involving awareness and control over attention) with lack of awareness and control over attention which can give rise to rumination and make it harder to break ruminative patterns.

-        Conclusion: How exactly did this meta-analysis “confirm the feasibility” of MBIs for rumination among patients with depression? This does not seem consistent with the scope of analysis conducted, which focused primarily on efficacy rather than feasibility.
